# Equine Arteritis Virus (EAV) Outbreak in a Show Stallion Population

**DOI:** 10.3390/v13112142

**Published:** 2021-10-24

**Authors:** Christiane Otzdorff, Julia Beckmann, Lutz S. Goehring

**Affiliations:** 1Equine Hospital, Center of Clinical Veterinary Medicine, Ludwig-Maximilians University, 85764 Oberschleißheim, Germany; Christiane.Otzdorff@gyn.vetmed.uni-muenchen.de; 2Equine Ambulatory Practice Dr. C. Krieg, 85656 Buch am Buchrain, Germany; Krieg@equine-vet.de; 3MH Gluck Equine Research Center, Department of Veterinary Science, College of Agriculture, Food & Environment, University of Kentucky, Lexington, KY 40546-0099, USA

**Keywords:** Equine viral arteritis (EVA), horizontal transmission, masturbate, venereal, respiratory tract, horse

## Abstract

(1) Background: Equine arteritis virus (EAV) infection causes reproductive losses and systemic vasculitis in susceptible *equidae*. The intact male becomes the virus’ reservoir upon EAV infection, as it causes a chronic-persistent infection of the accessory sex glands. Infected semen is the main source of virus transmission. (2) Here, we describe acute EAV infection and spread in a stallion population after introduction of new members to the group. (3) Conclusions: acute clinical signs, acute phase detection of antigen via (PCR) nasal swabs or (EDTA) blood, and seroconversion support the idea of transmission via seminal fluids into the respiratory tract(s) of others. This outbreak highlights EAV’s horizontal transmission via the respiratory tract. This route should be considered in a chronic-persistently infected herd, when seronegative animals are added to the group.

## 1. Introduction

Equine arteritis virus (EAV) is an enveloped, single-stranded, positive-sense RNA virus of the family *Arteriviridae* (genus *Alpharterivirus*, order *Nidovirales*). EAV infection is species-specific. The intact male, infected during adulthood, also known as a ‘stallion’, and not the adolescent intact male or ‘colt’ infected prior to or during puberty, plays an important role in the epidemiology of EAV. Upon infection, the stallion’s reproductive tract (specifically, the accessory sex glands) become persistently infected, and the stallion’s ejaculate becomes the main pathway of transmission. Seronegative mares inseminated with EAV-positive semen will not conceive, likely due to endometrial and vascular replication of virus. It is likely that these mares develop a systemic infection resulting in a generalized vasculitis, which will involve the respiratory tract, conjunctivae, and various mucous membranes. This phase, also known as Equine viral arteritis (EVA), is often associated with fever depending on infectious dose and strain differences, conjunctivitis, respiratory disease (nasal discharge, mandibular lymph node enlargement), and variable degrees of limb and/or ventral edema. Conjunctival, respiratory secretions, and saliva will contain high EAV titers and will contribute to horizontal aerosol or smear infection transmission into other members of the herd, however, with productive infection only in the EAV seronegative animals. Horizontal transmission is the alternative mode of virus spread among horses, and many clinical signs mimic those of Equid herpesvirus (EHV)-1 infection, Equine influenza, or *Streptococcus equi* spp. *equi* (strangles) infection in horses. EAV infection is characterized by rapid (horizontal) spread among seronegative animals.

In a herd of seronegative pregnant mares, a horizontal EAV spread will result in further reproductive losses. A vertical transmission has been described in late-stage pregnancies. Foals are born with a progressive and fulminant interstitial pneumonia and fibronecrotic enteritis. Scrotal enlargement in the intact male is a hallmark finding of EAV acute infection and is explained by fluid accumulation in the scrotal vaginal cavity. However, scrotal enlargement is not pathognomonic for EAV, as similar findings can be obtained during EHV-1 infection. In addition, the condition has to be differentiated from several local, regional, or systemic inflammatory conditions; from neoplastic disease; or from chronic heart failure [1,2].

As seropositive animals are protected against re-infection, and the virus is host-specific, its survival strategy focuses on a chronic-persistent infection state of the accessory sex glands in the stallion. Neither the castrated male, nor the (pre-) pubertal intact male, can enter a carrier phase, as it is apparently a testosterone-dependent process [3]. The carrier state can persist for months to years, and it can be life-long as well.

Two commercial vaccines, a modified-live vaccine and an inactivated vaccine, are available. As the two vaccines are licensed differently throughout the world, the inactivated version (Artervac, Zoetis, Berlin, Germany) is available in Germany and in several other countries of Europe. Both vaccines induce a robust immunity and claim to prevent carrier state induction in stallions. However, this claim of the inactivated vaccine is less well characterized.

The aim of this study is to raise awareness for EAV as an often-overlooked pathogen for respiratory tract disease with horizontal transmission. In addition, we want to raise awareness for transmission of urinary-genital fluids from shedding stallions as possible sources of respiratory tract infection in herd members.

## 2. Case Presentation

Six weeks prior to equine hospital admission at LMU Munich, a 12-year-old Pura Raza Espanola (PRE) stallion was added to a sizeable (mostly) stallion herd of about 50 animals. To be exact, because of COVID-19, a number of satellite operations in Germany, touring with show stallions, were combined at a central location in the Munich, Upper Bavaria area. All animals were housed in individual boxes/stalls, separated by metal bars, and spread out over four semi-connected stable units (A through D). Animals had daily exercise and trained in fixed groups. One stallion was found with a fever (rectal temperature of 40 °C (normal: 37.3–38.2 °C)), first noticed on a Tuesday in April 2020 (day 1 (D1)). A non-painful scrotal enlargement was detected by Thursday (D3). In addition to the persistent fever, this prompted a referral to the LMU Equine Hospital. The horse was quiet, alert, and responsive upon arrival. His respiratory rate was 16 breaths/minute (normal: 8–12); he was mildly tachycardic, with a heart rate of 52 beats/min (normal: 28–40). His oral mucous membrane color was bright pink, further moist, and without damages or visible petechial hemorrhage. Both mandibular lymph nodes were slightly enlarged and moderately painful on palpation. His rectal temperature upon arrival was 38.6 °C. Both scrota were enlarged, but they were not painful upon palpation (Figure 1). Auscultation of heart and lungs was within normal limits, and no unusual findings were detected during transrectal palpation of the abdomen. An ultrasound examination of the scrotum showed normal architecture of the testes and epididymis and an increased volume of an otherwise clear fluid in the vaginal cavity (hydrocele). Excess free fluid in the abdominal cavity could not be detected.

The horse was admitted and stabled, isolated from the hospital population, as this presentation in combination with a fever can be associated with EHV-1 or EAV infection. We received D2-laboratory results from the referring veterinarian, together with the results of our emergency laboratory. No gross abnormalities were noted on the D2 results. In-house results revealed a low normal leukocyte count (5.6 G/L, reference range 5–10 G/L) and a mildly increased acute phase protein serum amyloid A (SAA). The SAA concentration of 60 mg/L (normal: <10) indicated unspecific inflammation. His thrombocyte count was high with 340 G/L (reference range 120–180 G/L), and a blood smear was evaluated for *Anaplasma phagocytophilum*, which were not detected. During the night, the stallion showed mild (abdominal) discomfort. Feed was withheld, and his discomfort resolved with hand walking and an (iv) injection of a generic non-steroidal anti-inflammatory drug (flunixin meglumine 1.1 mg/kg, i.v. Flunidol, CP-Pharma, Burgdorf, Germany). The stallion remained afebrile following this injection for the remainder of his stay. On D4, we submitted one of two nasal swabs, collected during admission, for a multiplex polymerase chain reaction (PCR) ‘respiratory panel’ (EHV-1/-4, EI, EAV, *Streptococcus equi* spp. *equi*, IDEXX Laboratories, Ludwigsburg, Germany) and a serum sample for *Streptococcus equi* spp. *equi* antibody detection via an enzyme-linked immunosorbent assay (ELISA; Labor Böse, Harsum, Germany). The second nasal swab was submitted for an in-house EHV-1/-4 quantitative (q)PCR (Veterinary Virology, Ludwig-Maximilians University Munich, Germany), which returned negative within 4 h. During the evening hours of D4, the stallion developed prominent limb and preputial edema, which was addressed with cold-hosing and dexamethasone administration (Dexadresson 0.05 mg/kg, i.m. once daily (CP-Pharma, Burgdorf, Germany)) and continued over the weekend. The swelling on the limbs and prepuce had disappeared by D7. Due to a potentially contagious disease in this horse, we advised the owner to keep the stallion at our isolation facility, and we further advised the management of the show operation to continue monitoring rectal temperatures of all horses at their premises and install barrier precautions between units.

On D12, the facility reported a fever in a second stallion. This horse, a 10-year-old PRE stallion (Table 2: H12), showed a similar scrotal enlargement as was noticed in the index case (IC), a fever (39.8 °C), and mild mandibular lymphadenopathy. The neighboring horse, a 4-year-old PRE stallion (Table 2: H11), reportedly was ‘off’ for two days prior to the neighbor’s fever episode; however, he maintained a normal temperature. Interestingly, his mandibular lymph nodes were also slightly enlarged and sensitive to touch, and both horses were new additions to the herd (January 2020). They were already transferred in a separate stable wing without contact with others, including separate personnel. Interestingly, unit A was the barn wing directly adjacent to where the IC was stabled prior to hospital admission (Table 1).

As we, at that point of time, did not have a confirmed diagnosis in the hospitalized stallion, we collected additional nasal swabs and EDTA-blood from both horses and submitted them for in-house EHV-1/-4 qPCR testing. Swabs were analyzed on Monday (D13) and were both negative. In the meantime, the multiplex qPCR respiratory panel results, collected on D4, reported the presence of Equine Arteritis Virus (EAV) cDNA. A (stored) serum sample of the index case, collected on D2, was retrieved and submitted, combined with a second serum sample collected on D17 for an EAV virus neutralization (VN) assay (IDEXX, Ludwigsburg, Germany). The titer on D2 was <1:4, while there was seroconversion to 1:128 in the D17 sample (Table 2). On D23, blood was drawn from 10 more animals (Friesian, Nonius, Lipizzaner, Welsh pony). Seven animals had an anti-EAV titer reported between 1:64 and >1:256. While positive VN titers could have been the result of recent exposure, in the absence of any clinical signs until D23, this suggested chronic-persistent infection. The Welsh pony (H02) stallion and one Lipizzaner stallion (H08) were seronegative (Table 2). Serum samples of the horses H11 and H12, with an acute presentation and examined on D12 at the facility, were also included; however, they returned with a seronegative result (<1:4). Upon investigation, apparently the serum sample that was sent to the laboratory was a D12 sample (i.e., an ‘acute’ sample) and not a D23 sample. We reported the EAV-positive swab result of the IC to the head of the veterinary virology diagnostic laboratory, and the laboratory confirmed the presence of the EAV antigen (low abundance) in the nasal swab of H12, but not in H11.

## 3. Discussion

Here, we observed an EAV outbreak in a sizeable (mostly intact) male horse population of various breeds, which included PRE, Lipizzaner, Friesian, Cremello, Arabian horses, and members of the Nonius breed. Few horses/ponies of ‘other breeds’ were also present. Our collected data in summary, showed: (i) a high EAV seroprevalence in randomly selected herd members; (ii) two, possibly three, horses with acute clinical signs of an EAV infection; (iii) all three animals were seronegative for EAV at time point of acute clinical presentation; and (iv) evidence of a respiratory tract infection with EAV, as we detected the antigen in nasal swabs via PCR (IC and H12) and noticed slightly enlarged mandibular lymph nodes in three horses (IC, H11, H12). This all suggests that susceptible, seronegative horses were infected via droplet infection; however, the origin of ‘droplets’ remains undetermined. We believe that somewhere in this infection, chain droplets must have been generated from masturbation, seminal fluids, or possibly from urine of an intact male with venereal shedding. Shedding might have either led to aerosolization, smear, or fomite transmission into a seronegative animal, resulting in a respiratory tract infection. Whether the IC was a close neighbor to the original venereal shedder (Table 1) or within the same show group and exercising together remains unclear. Alternatively, the actual first horse infected by venereal fluids was not the 12-year-old PRE (IC), as we thought, but another horse where clinical signs were missed.

Nevertheless, the first route of transmission had to be from venereal shedding of a persistently infected stallion into the respiratory tract of a susceptible in-contact horse. Typically, case one of an outbreak is the seronegative mare that becomes infected during insemination with EAV-positive semen. Regardless, whether there are other (seronegative) mares inseminated with EAV-positive semen or there is respiratory droplet spread, pregnancy losses, or unexpected low pregnancy rates is a more robust and a more typical indicator of EAV infection [4]. However, infection transmission among male horses has been described before in a group of individually housed Lipizzaner stallions in South Africa [5]. Close contact between stallions is unusual; however, where group husbandry of a bachelor herd is practiced or stallions exercise as a group, these circumstances can be permissive for venereal transmission. These circumstances can initiate masturbation or mounting behavior, where pre-ejaculation fluids or urine containing EAV contaminates the direct environment.

In our IC, the scrotal swelling was caused by fluid accumulation in the vaginal cavity and noticed almost instantly with the onset of fever. Fluid build-up appeared not as a result of gravity accumulation, as there was no (visible) abdominal effusion. Vaginal cavity fluid accumulation in lieu of abdominal effusion is only possible if local vasculature is affected. We suspect that this is a targeted infection of male genital vasculature, possibly the pampiniform plexus system of the testicles [1,6].

## 4. Consequences for this Population

Confirmed EAV infection or seroconversion in the adult, intact male horse is a strong indicator for chronic-persistent infection resulting in venereal shedding. Persistent infection can last for weeks to years, but it can also be life-long. Persistent infection, when transmission results in reproductive loss in mares, and the semen of chronically infected males can only be used safely in seropositive mares. Interestingly, the chronic-persistent infection status is testosterone dependent. Seronegative geldings and pre-pubertal male horses will go through stages of a primary infection; they will seroconvert; however, they will not become persistently infected. Hence, castration is an effective measure of intervention in chronic shedders. However, this rather drastic measure also ends a career in reproduction and is a loss for future progeny. Few options for (temporary) castration have been explored in the past using chemical or immunological interventions, causing a temporary decrease in testosterone production with a variable effect. While this could mean a cessation of breeding activities for one or more breeding seasons, the hope is to clear the infection during periods of low or absent testosterone concentrations and return to normal breeding activities after cessation of these interventions. Any of these treatments definitively bear at least a temporary, if not permanent, risk for loss of libido, testicular function, and semen quality. However, temporary measures could bring back a successful stallion with genetical potential or keep a stallion available for a breed with a narrow genetical base. Alternatively, as effective vaccines against EAV are available to many parts of the world, mares that are wished to be bred to an EAV positive stallion can be vaccinated prior to breeding. These mares will conceive normally, and vertical transmission into the foal has not been observed.

Vaccination will not sanitize a chronic-persistent infection in a stallion. While the show horse herd described here was not intended for breeding, any behavioral changes, changes in hierarchy, or changes in demeanor, as evoked by (temporary) castration as described above, can have a huge impact on show performance and potentially also on the safety for employees or others. While we were told that none of these horses is ever sold or given away when retired and that animals stay at special retirement facilities until death, the risk for spread of disease is through breeding accidents, clandestine breeding, or through the addition of seronegative members to the group. It is important that this is discussed with the owners, pointing out their responsibilities. Vaccination of new (seronegative) additions to the herd can prevent flare-ups of acute disease, and it has to follow a stringent protocol of testing and (re)vaccination. Nevertheless, EAV is an ongoing threat to the equine population worldwide. EAV has been detected in many breeds. An ever-increasing showcase of stallion groups touring from place to place that are not used for breeding increases the risk of disease spread in the equine population. Currently, a seroprevalence of 10–20% of (continental) Europe’s horse population requires more frequent testing and reporting for a better idea of disease prevalence and dynamics.

## Figures and Tables

**Figure 1 viruses-13-02142-f001:**
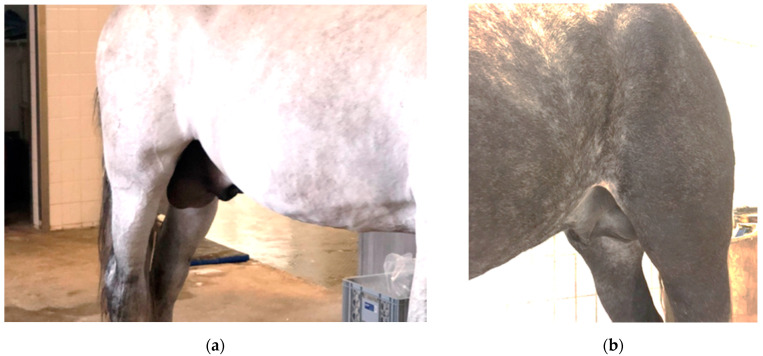
(**a**) extensive scrotal swelling in this 12-year-old PRE stallion (index case); (**b**) age- and size-matched control (9-year-old draft horse (Percheron) stallion). Furthermore, note the (subjectively) extensive distention of peripheral blood vessels of the chest and limb in the index case.

**Table 1 viruses-13-02142-t001:** Floor plan of stabled horses and ponies at the facility, all males were intact, except where indicated as ‘gelding’, and spread over four subunits (A–D). In yellow: horses with clinical signs; in green: additional serum samples for the anti-EAV antibody titer (virus neutralization VN); see Table 1. Symbols: (○), seronegative; (●), seropositive; (○) horseID > (●), seroconversion; (◎), free dressage training as a group (unit B).

**Unit A**	**Entrance**	
(Shetland pony)	**aisle**	
(Shetland pony)		

○ H12 (PRE)		
○ H11 (PRE)		
H02 (Welsh pony)		
**tack room**		**wash stall and grooming**
**unit B**	**hallway**	
H04 (Friesian) ● ◎	**aisle**	○ IC (PRE) >●
H03 (Friesian) ● ◎		H10 (Friesian) ● ◎
H01 (PRE) ● ◎		(Warmblood) ◎
(Varner) ◎		(Draft horse) ◎
**unit C**	**hallway**	
H05 (Nonius) ●	**aisle**	H09 (Nonius) ●
(Nonius, gelding)		H06 (Nonius) ●
(Nonius)		H13 (Nonius) ●
(Nonius)		H07 (Nonius) ●
(Lusitano)		(Nonius)
H08 (Lipizzaner) ○		(Nonius)
staircase		(Lusitano)
**unit D**	**hallway**	
(Lusitano)	**aisle**	(Lusitano)
(Lusitano)		(Lusitano)
(Arabian)		(Lusitano)
(Arabian, gelding)		(PRE)
(Arabian)		(Ibero)
(Arabian)		(Ibero)
(Arabian)		(Ibero)
(Arabian)		(Ibero)
(Arabian)		(Ibero)
(Arabian)		(Ibero)
(Arabian)		(Ibero)
	**entrance**	

**Table 2 viruses-13-02142-t002:** Serology data on select horses.

	D2	D12	D17	D23
Index Case	<1:4		1:128	
H 01				1:128
H 02		no sample		
H 03				>1:256
H 04				>1:256
H 05				1:128
H 06				1:128
H 07				1:128
H 08		<1:4		
H 09				1:64
H 10		1:12		
H 11		<1:4		<1:4
H 12		<1:4		
H 13				1:128

## Data Availability

Not applicable.

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
