# Peer review of "Equine Arteritis Virus (EAV) Outbreak in a Show Stallion Population"

_viruses, 2021, doi:10.3390/v13112142_

Round 1

Reviewer 1 Report

Authors have described an outbreak of Equine Arteris Virus in a group of mixed population horses stabled together.  Based on the information gleaned from random testing, there appears to be spread between horses that suggests respiratory transmission could have played a large role. This is an important observation as I get the impression (this is not my area) that horse breeders do not consider aerosol spread a major concern for this virus.. Given how infectious this virus is within horse populations, this should be given more attention.  Authors could beef up the language if the belief is that aerosol transmission is not given the attention it deserves. There is a vaccine available, I think given this concern of spread through aerosol/droplets, authors could expand discussion / conclusion to make a case that vaccines should be required for all horses / mandatory when being sold and moved into new herd situations.

Author Response

Dear reviewers,

We thank the reviewers for their time spent carefully reviewing the manuscript, and in their opinions regarding this case report. We greatly appreciate the thorough and thoughtful comments provided on our submitted article.

Reviewer 1:

Authors have described an outbreak of Equine Arteris Virus in a group of mixed population horses stabled together.  Based on the information gleaned from random testing, there appears to be spread between horses that suggests respiratory transmission could have played a large role. This is an important observation as I get the impression (this is not my area) that horse breeders do not consider aerosol spread a major concern for this virus.. Given how infectious this virus is within horse populations, this should be given more attention.  Authors could beef up the language if the belief is that aerosol transmission is not given the attention it deserves. There is a vaccine available, I think given this concern of spread through aerosol/droplets, authors could expand discussion / conclusion to make a case that vaccines should be required for all horses / mandatory when being sold and moved into new herd situations.

Dear reviewer, thank you very much for your valuable comments. We reorganized the introduction and added to the pathogenesis of this virus and on vaccination strategies that are currently implemented (differently worldwide). As EAV infection has a high morbidity/low mortality in the horse and infection provides a robust immunity most of the owners will take the risk of once-in-a-lifetime infection rather than biannual vaccination.

Thank you for your valuable time!

Reviewer 2 Report

General comment

The disease is called Equine Viral Arteritis (EVA) and the causative agent is Equine Arteritis Virus (EAV), please correct this in the manuscript (Udeni and Balasuriya, 2014).

What about vaccination against EVA in Germany?

Specific comments

Line 12 – intact male.

Line 14 – delete the space before the number (2).

Please add keywords.

Line 26 – fever instead of fevers.

Line 34 – please add a dot after the (1,2).

Line 38 – “intact male adult horse”, please change to adult intact male horse or maybe use the word stallion.

Lines 55-56 – please be consistent (D)1 or D1 (like D3).

Line 81 – please write the full term of NSAID.

Line 83 – please write the full name of S. equi.

Line 83 – name of bacteria should be written in Italics.

Line 84 – please write the full meaning of the word ELISA.

Line 88 – please include dose and route of drug administration.

Line 104 and 113 – please write the full term of VN.

Lines 128-130 - I think that number below 10 should be written in words.

Line 163 – please add a dote at the end of the sentence.

Author Response

Dear reviewers,

We thank the reviewers for their time spent carefully reviewing the manuscript, and in their opinions regarding this case report. We greatly appreciate the thorough and thoughtful comments provided on our submitted article.

General comment

The disease is called Equine Viral Arteritis (EVA) and the causative agent is Equine Arteritis Virus (EAV), please correct this in the manuscript (Udeni and Balasuriya, 2014).

What about vaccination against EVA in Germany?

We applied the two abbreviations throughout the text where appropriate, reorganized the introduction, and included a section for strategic vaccination with available vaccines.

Specific comments

Line 12 – intact male. We have changes it to “intact male”.

Line 14 – delete the space before the number (2). We have removed the space.

Please add keywords.

We thank the reviewer for pointing that out. As EAV (Equine arteritis virus) is already part of the title; we added the following keywords: Equine viral arteritis (EVA), horizontal transmission, venereal, masturbate, respiratory tract, and horse

Line 26 – fever instead of fevers. We corrected.

Line 34 – please add a dot after the (1,2). We corrected the dot.

Line 38 – “intact male adult horse”, please change to adult intact male horse or maybe use the word stallion. Thank you for your comment. We expanded the introduction and introduced the descriptors for adolescent (colt) and adult (stallion) intact male.

Lines 55-56 – please be consistent (D)1 or D1 (like D3). Thank you for pointing that out. We introduced the abbreviation first as day 1 (D1), then used D1, D3, etc  throughout the manuscript.

Line 81 – please write the full term of NSAID. As this is a group of drugs and commonly abbreviated as NSAID we switched to ‘a generic non-steroidal anti-inflammatory drug’ and added the product (flunixin meglumine at its administered dose and route).

Line 83 – please write the full name of S. equi. Name of bacteria should be written in Italics. Thank you very much for pointing that out. We changed to Streptococcus equi spp. equi

Line 84 – please write the full meaning of the word ELISA. We changed the term ELISA to the full meaning Enzyme-linked Immunoabsorbent Assay).

Line 88 – please include dose and route of drug administration.

Thank you for your comment. Of course, we added the dose for dexamethasone (0.05 mg/kg) and the route of administration (i.m.).

Line 104 and 113 – please write the full term of VN. We clarified this and it is now written “virus neutralization” assay.

Lines 128-130 - I think that number below 10 should be written in words. Thank you for pointing that out. However, this is a journal’s choice, and we are required write numbers to save space. Exception is at the very beginning of a sentence.

Line 163 – please add a dote at the end of the sentence. Please excuse the mistake. We added.

Thank you for your valuable time!

Reviewer 3 Report

The manuscript by Otzdorff et al. entitled "Equine Arteritis Virus outbreak in a stallion (show horse) population," describes the clinical presentations and diagnostic investigation of an EAV outbreak in show horses. Overall, manuscript does not provide much new scientific insight. However, as there are a limited number of publications available on the topic, the manuscript has merit and should be considered for publication. There are, however, a number of issues that should be addressed.

  1. General comment: the manuscript will requires substantial English language edits. Also please change "mandibular lymph node" to "submandibular" lymph node throughout the manuscript.
  2. Abstract: 
    • Line 14: Please delete "used for insemination".
    • Line 18-19: The sentence is incomplete. Please reword.
  3. Introduction:
    • Line 25-28: Please change "which results" to "might result" in high fever. While strains with high pathogenic potential might cause all of the symptoms listed, infection with some strains will result in mild or no clinical signs.
    • Line 30: Please change the sentence to "seronegative pregnant female".
    • Lines 41 and 42: Please delete parentheses.
  4. Case presentation:
    • Line 59: Please change to "His oral mucous membrane color..."
    • Lines 75 and 76: Please provide reference ranges.
    • Line 78-79: Please delete the sentence.
    • Line 81: Please state what NSAID was given and provide the dosage.
    • Line 88: Please provide the dosage, route, and frequency of administration. 
    • Line 89: Please specify which swelling had disappeared.
    • Line 91: please change "body" to "rectal" temperature and remove the parenthesis.
    • Paragraph starting at line 93: This paragraph is difficult to follow, as the authors are inconsistent in the designations used for various horses (e.g. line 96: while the body of the manuscript states "The neighboring horse, a 4-year-old PRE stallion [H11]..." the figure provide suggests that H11 was a Welsh pony).
    • Paragraph starting at line 107: Same comment as above. Please use letter/number designations to consistently identify animals.
    • Line 113: Please define VN.
  5. Discussion:
    • Line 127: While several horses were seropositve upon testing, their serological status was unknown prior to the outbreak. Thus, it is unclear whether these horses were infected during the described outbreak. Please add a statement addressing this matter.
    • Line 132: Please replace the word "receptive" with "susceptible".
    • Line 157-161: This statement is purely speculative. What about peripheral limb edema seen in this horse?

Author Response

The manuscript by Otzdorff et al. entitled "Equine Arteritis Virus outbreak in a stallion (show horse) population," describes the clinical presentations and diagnostic investigation of an EAV outbreak in show horses. Overall, manuscript does not provide much new scientific insight. However, as there are a limited number of publications available on the topic, the manuscript has merit and should be considered for publication. There are, however, a number of issues that should be addressed.

  1. General comment: the manuscript will requires substantial English language edits. Also please change "mandibular lymph node" to "submandibular" lymph node throughout the manuscript.

We appreciate the reviewer’s suggestion. However, on several occasions in earlier publications these authors were alerted by reviewers or editors that the correct nomenclature defined by Acta Anatomica  is “mandibular lymph node” or Lnn. mandibulares.

  1. Abstract: 
  • Line 14: Please delete "used for insemination".

Thank you very much for your comment. We agree and deleted “used for insemination”.

  • Line 18-19: The sentence is incomplete. Please reword.
  1. Introduction:
  • Line 25-28: Please change "which results" to "might result" in high fever. While strains with high pathogenic potential might cause all of the symptoms listed, infection with some strains will result in mild or no clinical signs.

Thank you very much for pointing that out. We changed the introduction as requested by a total of two reviewers. For the clinical signs we changed to ‘This phase, also known as Equine viral arteritis (EVA) is often associated with fever depending on infectious dose and strain differences; conjunctivitis; respiratory disease (nasal discharge, mandibular lymph node enlargement); variable degrees of limb and/or ventral edema.‘

  • Line 30: Please change the sentence to "seronegative pregnant female".

We fully agree and added the term “seronegative”.

  • Lines 41 and 42: Please delete parentheses. We did on both occasions.

  1. Case presentation:
  • Line 59: Please change to "His oral mucous membrane color..."

We agree and exchanged the word “mouth” with ‘oral’

  • Lines 75 and 76: Please provide reference ranges.

We apologize for our negligence of not mention the reference ranges. We added these to all parameters.

  • Line 78-79: Please delete the sentence.

Thank you for your comment. We totally agree and deleted the sentence.

  • Line 81: Please state what NSAID was given and provide the dosage.
  • Line 88: Please provide the dosage, route, and frequency of administration. 

Thank you very much for pointing this out. We apologize for the absence of this important information and added it to the NSAID and the dexamethasone treatment.

  • Line 89: Please specify which swelling had disappeared.

We added additional information. The sentence was changed to “The swelling on limbs and prepuce had disappeared by D7.”

  • Line 91: please change "body" to "rectal" temperature and remove the parenthesis.

We changed “body” to “rectal” temperature.

  • Paragraph starting at line 93: This paragraph is difficult to follow, as the authors are inconsistent in the designations used for various horses (e.g. line 96: while the body of the manuscript states "The neighboring horse, a 4-year-old PRE stallion [H11]..." the figure provide suggests that H11 was a Welsh pony).

Please excuse the mistake and the confusion regarding the designation of the various horses. We changed the horse IDs in the text and in the figure.

  • Paragraph starting at line 107: Same comment as above. Please use letter/number designations to consistently identify animals. We corrected this discrepancy.
  • Line 113: Please define VN. We changed the abbreviation to “virus neutralization”.
  1. Discussion:
  • Line 127: While several horses were seropositve upon testing, their serological status was unknown prior to the outbreak. Thus, it is unclear whether these horses were infected during the described outbreak. Please add a statement addressing this matter. You are certainly correct, and we added the following sentence: While positive VN titers could have been the result of recent exposure, in the absence of any clinical signs until D23 this suggests chronic-persistent infection.
  • Line 132: Please replace the word "receptive" with "susceptible".

We agree and wrote “susceptible” instead of “receptive”.

  • Line 157-161: This statement is purely speculative. What about peripheral limb edema seen in this horse? We were commenting on the sequence of events, and painful limb edema as a result of vasculitis in the IC developed after fever broke and several days after the scrotal swelling was noticed. Therefore, we believe there is a possible predilection for the vasculature of the male gonads, followed by limb edema that began several days later. We agree, it is speculative, and we reworded this section to: In our IC the scrotal swelling was caused by fluid accumulation in the vaginal cavity and noticed almost instantly with the onset of fever. Fluid build-up appeared  not a result of gravity accumulation as there was no (visible)abdominal effusion. Vaginal cavity fluid accumulation in lieu of abdominal effusion is only possible if local vasculature is affected. We suspect, that this is a targeted infection of male genital vasculature, possibly the pampiniform plexus system of the testicles.

Once again, we thank you for your valuable time you put in reviewing our paper and look forward to meeting your expectations.

Round 2

Reviewer 2 Report

I apologize but I am still not convinced that there is enough data to support an outbreak report. Although the index case did suffer from EVA (based on PCR and serum conversion), acute infection with EVA was not diagnosed in any other horse (except in a single seropositive result in few horses). Please correct me if I am wrong. Indeed the high seroprevalence (which does not indicate acute infection) was found in the sampled horses. But this was not found in the horses that did show clinical signs and were housed in a different unit from the IC. Maybe if you could demonstrate that the seroprevalence in other horses (for example in unit D) that had no physical contact with the other suspected cases was lower that could support the claim. 

Author Response

Dear Reviewer, 

thank you for your valuable time in this review process. It is greatly appreciated. We fully understand your concerns. The lack of proof of infection with EAV came because of a miscommunication with the on-site veterinarian. To save costs serum samples which should have provided proof of seroconversion were not collected when other horses were sampled to detect seroprevalence. However, we have an EAV positive nasal swab on H12 on the day we examined the horse because of a fever and scrotal enlargement. The nasal swab was collected to rule out EHV-1 at that time; however, when I mentioned the EAV diagnosis in the index case to the head of the virology diagnostic lab, he ran an additional EAV PCR out of own interest on the sample, and it was positive at CT 34 for EAV antigen.  H11 and H12 were both seronegative on the day of sampling, and both horses were also recent additions to the herd, similar to the index case. I hope you agree that this is proof of spread within the group of seronegative new additions of the herd with involvement of the respiratory tract. I will add these findings to the text . 

I can get you in contact with the head of virology if you need further confirmation. Thank you very for pointing out this discrepancy in the text.